# Aqueous Solutions of Peptides and Trialkylamines Lead to Unexpected Peptide Modification

**DOI:** 10.3390/molecules26216481

**Published:** 2021-10-27

**Authors:** Yiran Ma, Puja J. Gandhi, James P. Reilly

**Affiliations:** Department of Chemistry, Indiana University, Bloomington, IN 47405, USA; yirma@iu.edu (Y.M.); gandhi.pj@gmail.com (P.J.G.)

**Keywords:** triethylamine, Schiff base, protein modification

## Abstract

When trialkylamines are added to buffered solutions of peptides, unexpected adducts can be formed. These adducts correspond to Schiff base products. The source of the reaction is the unexpected presence of aldehydes in amines. The aldehydes can be detected in a few ways. Most importantly, they can lead to unanticipated results in proteomics experiments. Their undesirable effects can be minimized through the addition of other amines.

## 1. Introduction

Trialkylamines are strong organic bases that have found numerous applications in bioanalytical chemistry. Triethylamine has been a traditional solvent or additive employed in chromatography systems during biomolecule analysis, as it enables better separations for ionic samples in both normal and reverse-phase liquid chromatography [1,2,3,4,5]. Triethylamine is also used in charge stripping processes in both the gas and liquid phases [6,7,8]. Gaseous triethylamine has been added to deprotonate cytochrome c dimers so that dimer peaks no longer overlap with monomer peaks [6]. Similarly, triethylamine was added to polyethylene glycol or PEGylated proteins in order to simplify their mass spectra [7].

In addition to its popularity in liquid chromatography systems, triethylamine has also been utilized to improve fluorescence sensors. Wang used triethylamine to deprotonate 2D covalent organic framework (COF) TpPa-1 so that the COF sensor improved 70-fold [9]. With the help of the improved fluorescence sensor, 117.5 nM methylglyoxal, which is a biomarker for diabetes mellitus diagnosis, could be detected [9]. Triethylamine has also been used as a high pH buffer for eluting proteins [10]. Due to its high pKa, it is also added to buffers to adjust pH [11].

Trialkylamines are traditionally prepared by reacting alcohols with ammonia, where noble metals are used as catalysts [12]. For example, the two reactants may be preheated under hydrogen before being passed through reduced copper and nickel catalysts at high temperatures [13]. Ammonia is then alkylated and forms saturated and unsaturated amines. The mixture is then separated by distillation [13]. The commonly known impurities from commercially available trialkylamines are water, ammonia and the unsaturated alkylamines [12].

When trialkylamines are added to biochemical samples, it is usually assumed that they do not modify proteins or peptides. As illustrated in the present work, this assumption is not necessarily true.

## 2. Results and Discussion

### 2.1. Analysis of Modified Hemoglobin Digest

During hemoglobin peptide digest labeling experiments using various reagents, the pH of buffer solutions was raised using triethylamine and several peptide masses were unexpectedly found to be shifted by +26 Da. It was quickly established that the 26 Da adducts could be observed simply by adding triethylamine to the aqueous peptide solution. To verify the origin of this effect, hemoglobin was digested and then incubated with triethylamine before being analyzed by MALDI. Mass spectrum of the control sample was recorded and shown in Figure 1a. All the observed peaks corresponded to expected, unmodified peptide masses. The MALDI spectrum for the modified hemoglobin digested peptides appears in Figure 1b. This experiment was also repeated with trimethylamine and tripropylamine, and these led to shifts of +12 Da and +40 Da, respectively. (Examples of tripropylamine reaction with angiotensin appear later in Figure 8). To verify that these adduct peaks were not any kind of MALDI artifacts, reaction products were also analyzed by ESI-MS. In addition, since these results were rather surprising, experiments were repeated with newly purchased trialkylamines and freshly distilled trialkylamines. In all cases, similar MALDI and ESI mass spectra and mass shifts were observed. The observed mass shifts were close to those expected for Schiff base reactions that occur between amines and aldehydes. Formaldehyde, acetaldehyde and propionaldehyde are known to yield mass shifts of 12, 26 and 40 Da [14], as illustrated in Figure 2. Nevertheless, it was surprising to us that trialkylamines might contain aldehydes that would enable the formation of these Schiff base products.

To determine where on the peptides the adducts were located, we fragmented several modified peptides when recording ESI-MS spectra using collision-induced dissociation (CID) on a Thermo LTQ ion-trap instrument. One example is shown in Figure 3. The digested peptide LLVVYPWTQR has the most intense peak in the MALDI spectrum, at 1274.8 Da. After being incubated with triethylamine, this peptide increased in mass by 26 Da, suggesting that one site had been modified. In the MS2 spectrum, the y_5_ to y_8_ ions were not shifted in mass, while b_2_ to b_5_ ions were all shifted, indicating that the adduct was within the first two residues of the N-terminus. Since these two residues were both leucine, we concluded that the reaction must have occurred at the N-terminal amine.

Another peptide, VNDEVGGEALGR, which has a precursor mass of 1314.6 Da, was fragmented by CID, and the resulting spectrum is displayed in Figure 4a. This is also one of the masses that can be found in the MALDI spectrum due to its basic arginine residue. Following exposure to triethylamine, the peptide mass shifted to 1340.7 Da, indicating that one site on the peptide was modified by 26 Da. The CID spectrum of this mass is shown in Figure 4b. It is evident from comparison of the two spectra that none of the observed y-type ions (y_2_–y_11_) were modified. Since the largest y ion observed was y_11_, the change must have occurred at the first two residues near the N terminus. This is consistent with all b ions that appear in the spectrum (b_2_–b_12_) being modified, again indicating that the modification involves the first two residues near the N-terminus. Since the side chains of both valine and asparagine are not reactive, we conclude that the modification took place at the N-terminus of this peptide.

The next peptide is FLASVSTVLTSK, as shown in Figure 5a. It has a mass of 1252.7 Da that is shifted by 26 Da after incubation. Similar to the previous two peptides, a modified b_2_ ion and unmodified y_10_ ions were found in the CID fragmentation spectrum as shown in Figure 5b. This suggests that the modification happened on the first two residues near the N terminus. Phenylalanine and leucine sidechains are not as reactive as the N terminus, thus making the primary amine on N-terminus the most likely reaction site.

### 2.2. Confirmation of Chemical Composition of Modification

To accurately measure our observed mass shifts, human angiotensin II was reacted with triethylamine for 2 h at 37 °C, and the product was analyzed with a Thermo orbitrap. We observed a mass shift of 26.016 Da for modified angiotensin. The theoretical mass shift for a Schiff base reaction between the peptide and acetaldehyde is 26.016 Da due to addition of C_2_H_2_. No other combination of atoms provides as good of a mass match. The next closest composition is CN, which has a mass of 26.0031 Da. Similarly, exposure of angiotensin to tripropylamine yielded a mass shift of 40.028 Da. The Schiff base adduct, C_3_H_4_, would lead to a mass shift of 40.031 Da. The next closest mass shift alternative of 40.019 Da would be for C_2_H_2_N.

### 2.3. Verification of Presence of Aldehydes in Trialkylamines

An attempt was made to detect aldehydes in trialkylamines using NMR, but none were observed. However, due to their different numbers of hydrogens, NMR is less sensitive to propionaldehyde than to tripropylamine, so it is not optimal for detecting small aldehyde impurities.

A more sensitive method, headspace GC-MS, was then employed to look for aldehydes in freshly purchased trialkylamines. Fractionational distillation of tripropylamine was performed to eliminate propionaldehyde. The total ion chromatogram of the distilled tripropylamine is illustrated in Figure 6. The total ion chromatogram indicated that the amount of propionaldehyde in tripropylamine decreased after distillation, but it did not completely disappear. This suggests that there is a trace amount of propionaldehyde present in commercially available tripropylamine. Other than in trialkylamine production, copper is also a useful catalyst in other reactions, such as alcohol oxidation and dehydrogenation [15,16]. Therefore, it is possible that copper assists in the formation of aldehydes during the trialkylamine production process. This could explain why propionaldehyde was found in tripropylamine, despite that it is not a primary product from tripropylamine synthesis. Propanol could have gone through dehydrogenation, leading to propionaldehyde.

To see whether the amount of aldehyde in tripropylamine increases after tripropylamine is in contact with water, the following experiments were conducted. Tripropylamine was incubated with water for 0 min, 2 h and 12 h at 37 °C before being analyzed by headspace GC-MS to monitor the amount of propionaldehyde in a tripropylamine water mixture without disturbing the vapor pressure of propionaldehyde. Since the boiling point of tripropylamine is much higher than 70 °C, the amount of tripropylamine vapor should stay constant in all three samples. Therefore, we can estimate the relative amount of propionaldehyde based on the areas for both tripropylamine peaks and propionaldehyde peaks in the total ion chromatogram. The total ion chromatograms for the three samples are shown in Figure 7. When the sample did not incubate before injection, the area for the propionaldehyde peak was 18.44% of the area for the tripropylamine peak. When the mixture was incubated for 2h before injection, the area for propionaldehyde peak increased to 28.16% of the tripropylamine peak. After 12 h of incubation, the propionaldehyde area climbed to 37.97% of the tripropylamine peak. Although the exact amount of propionaldehyde cannot be determined, we can still conclude that propionaldehyde concentration doubled after 12 h of incubation time. It is unclear exactly how aldehydes are generated by alkylamine contact with water. However, it is evidently a slow oxidation process involving air or water.

Another test that was applied to verify the existence of aldehydes was the Schiff test. This is a common method that is used to qualitatively detect aldehydes in samples [17]. Adding Schiff’s reagent to a solution that contains aldehydes in an acidic environment leads to a purple or pink color. Unfortunately, since the Schiff reagent contains water, we were unable to examine whether there were any aldehydes in trialkylamines in non-aqueous conditions.

When the triethylamine: HCl molar ratio was set to 1.24:1, the solution indeed turned purple and then returned clear when the Schiff’s reagent was added, consistent with aldehydes being present in the solution [18,19]. However, the Schiff test yielded no quantitative information. This is because it is difficult to match the pH of trialkylamine solutions with the aldehyde standards. A small variance in pH considerably affects the maximum absorbance due to the three protonation states of Schiff’s reagent. When the solution is too acidic, Schiff’s reagent does not change color because a fully protonated Schiff’s reagent is clear. When the solution is too basic, the solution does not respond to the aldehyde concentration.

### 2.4. Critical Conditions for Trialkylamine Modifications

Several experiments were performed to understand the conditions that were important for reaction of aldehydes with peptides. In one study, 1.70 mM propionaldehyde was mixed with 50 μM angiotensin in water, and, as expected, a reaction occurred to form a 40 Da adduct (Figure 8a). When anhydrous DMSO was used instead of water, propionaldehyde reacted with angiotensin and yielded a more intense adduct signal (Figure 8b). This indicates that an aqueous environment is not necessary for the reaction between aldehydes and peptides to form Schiff bases. On the other hand, when 333 mM ammonium bicarbonate at pH 8 was used as a buffer, the reaction between propionaldehyde and angiotensin yielded substantially more adduct compared with aqueous solvents, implying that the reaction between angiotensin and aldehydes was accelerated by more basic conditions (Figure 8c). This is consistent with previous studies [20,21]. It should be noted that propionaldehyde concentrations were the same for all three experiments.

To test whether an aqueous environment is critical for tripropylamine and peptide reactions, 3.20 M tripropylamine was mixed with 50 μM angiotensin either in water or anhydrous DMSO. The solutions were mixed and left at 37 °C for 2 h. MALDI-TOF spectra showed that significant amounts of adducts were formed in both solvents (Figure 8d,e).

In conclusion, water was not necessary for tripropylamine to react with peptides to form Schiff base adducts. The amount of adduct obtained with 1.70 mM propionaldehyde was similar to that with 3.20 M tripropylamine in DMSO. This suggests that the percentage of propionaldehyde in our tripropylamine is about 0.05%. This is within the 2% overall impurity limit claimed by the manufacturer. In contrast, 1.70 mM propionaldehyde did not yield the same amount of adduct as 3.20 M tripropylamine with angiotensin in water, as shown in Figure 7a,d; this is likely due to the different pHs of these solutions. It is also possible that the water and tripropylamine generated additional propionaldehyde during incubation.

### 2.5. Elimination of Undesirable Aldehyde-Induced Modifications with Propylamine

Our next experiments aimed to determine whether we could block the reaction of aldehydes with peptides. Since n-propylamine has a primary amine, we expected that it would be a good blocking candidate. A measure of 12 μmol n-propylamine was added to 70 nmol propionaldehyde in DMSO and incubated for 20 min before 2 nmol of angiotensin was added. The solution was then vortexed and left at 37 °C for 2 h before being dried and analyzed by MALDI. A mass spectrum showing the results of this reaction is displayed in Figure 8f. The lack of an adduct demonstrates that the propylamine had indeed blocked the reaction of angiotensin with propionaldehyde.

To prevent peptide modifications that were occurring in trialkylamine solutions, propylamine was introduced. First, 12 μmol n-propylamine were mixed with 179 μmol triethylamine or 131 μmol tripropylamine with water and left at room temperature for 20 min. Next, 2 nmol of angiotensin was added to the mixture. The final concentrations of trialkylamines and angiotensin were the same as in all previous experiments. No trialkylamine-peptide reaction was observed in water or in anhydrous DMSO. Mass spectra looked essentially identical to that displayed in Figure 8f.

## 3. Materials and Methods

### 3.1. Materials

Human hemoglobin, Angiotensin II, trimethylamine, tripropylamine, propylamine and Schiff’s reagent were purchased from Sigma-Aldrich. MS-grade trypsin was obtained from Thermo scientific (Rockford, IL, USA). Triethylamine and 3 Å Molecular sieves (8–12 mesh beads) were procured from EMD Millipore (Darmstadt, Germany). Acetaldehyde and propionaldehyde were purchased from TCI America (Portland, OR, USA).

### 3.2. Methods

To begin, 25 μg of hemoglobin was digested with 1μg trypsin overnight at 37 °C in an ammonium bicarbonate buffer at pH 8 and subsequently lyophilized in a SpeedVac (Jouan, Winchester, VA, USA). For control experiments, digested peptides were directly re-dissolved in water and spotted on a MALDI plate. A measure of 0.65 μL of 10 g/L CHCA (50:50 water to ACN with 0.1% TFA) matrix was added on the same spot after peptide had dried. MALDI spectra were taken with the 4800 MALDI TOF-TOF analyzer. For experiments with trialkylamines, the digested hemoglobin peptides were suspended in 14 μL of water and 25 μL of triethylamine. The solution was vortexed for 30 min at 37 °C before being lyophilized again. The dried product was re-dissolved in water and spotted as above on a MALDI plate as described before.

For ESI-MS experiments, samples were prepared in a similar manner. Digested hemoglobin peptides were injected into an Eksigent NanoLC-2D and eluted into a Thermo LTQ ion trap mass spectrometer. A solution of 0.1% formic acid was used as solvent A, and 0.1% formic acid in ACN was used as solvent B to produce the LC gradient. A 150 min gradient at 300 nL/min separated the peptides. Solvent B increased from 5% to 35% during 1 h and then raised to 90% over 40 min. Solvent B was then held at 90% for 20 min before dropping to 5% again in 25 min. Solvent B was kept at 5% for 5 min at the end of the gradient. A home packed 200 Å C18 column was used for trapping, and a 120 Å C18 column was used for separation. The loose packing media were both purchased from Bischoff. Both columns were 75 μm ID. The trapping and analytical columns were 2 and 10 cm, respectively. The CID energy employed in the fragmentation method was 35%. Resulting raw files were converted to MGF files and analyzed by Protein Prospector.

For more accurate mass measurements, 50 μM angiotensin reacted with 4.37 M triethylamine or 3.2 M tripropylamine in water at 37 °C for 2 h. The modified peptides were then dried in a speedvac and reconstituted in water for MS analysis. A Thermo Fusion Lumos Tribrid mass spectrometer coupled with Thermo Scientific EASY nLC 1200 was employed in these experiments. The sample went through a 15 min gradient at 300 nL/min, where solvent A was 0.1% formic acid in water and solvent B was 0.1% formic acid in 80% ACN. The gradient started at 2% solvent B and increased to 8% solvent B over the course of 1 min, then moved to 40% solvent B in the next 9 min. The composition of solvent B was then sharply increased to 100% in 1 min and maintained for the last 4 min. The gradient was short because the modified peptides did not require extensive separation. The chromatography was conducted at room temperature. The trapping column was a Thermo Acclaim PepMap 100 (75 μm × 2 cm), and the analytical column was a 25 cm version of the same material.

^1^H NMR spectra were taken with a Varian 400 MHz Inova spectrometer.

Headspace experiments were performed with an Agilent 7890B/7250 GC-QTOF. In this work, 0.5 mL of pure tripropylamine was placed in a 10 mL bottle and heated to 70 °C for 7.5 min before 1 mL of the vapor above the liquid was injected into a GC-MS for mass measurement. Tripropylamine and propionaldehyde have boiling points of 156 °C and 49 °C respectively. By heating the tripropylamine solution to 70 °C, injection of aldehyde vapor into the GC-MS was favored, facilitating its detection.

For Schiff’s test standards, the solubility of water in triethylamine is very limited. During peptide experiments, the peptides were modified despite triethylamine not being completely miscible with water. When using Schiff’s reagent, however, it is important that the solution is aqueous. Fortunately, triethylamine is protonated by the addition of acid, and this forms a homogenous solution with Schiff’s reagent. HCl was added to achieve a neutral or acidic pH, which was tested with pH paper, before Schiff’s reagent was added. A measure of 12 μL to 96 μL of acetaldehyde was diluted with water to make a total volume of 1262 μL. An amount of 140 μL Schiff reagent was then added to the diluted acetaldehyde solution to make a standard series ranging from 0.766 mM to 6.10 mM. The standards were added to a cuvette and the absorbance was recorded with a Thermo Scientific Genesys 10S UV-Vis spectrophotometer. The wavelength at maximum absorbance was found to be 558 nm and absorbance for all solutions were recorded at this wavelength. For triethylamine samples, 6M HCl was used to acidify the solution. A range of 600–705 μL of HCl was added to triethylamine to make a final volume of 1262 μL. The molar ratio between triethylamine and HCl ranged from 1.05:1 to 1.32:1. A measure of 140 μL of Schiff’s reagent was then added to the mixture.

## 4. Conclusions

This study aimed to investigate peptide modifications caused by alkylamines. Peptide mass shifts were found to be due to Schiff base reactions and were confirmed by accurate mass measurements. The existence of aldehydes in trialkylamine was also verified with headspace GC-MS and Schiff’s test. Various conditions were tested for Schiff base reaction and tripropylamine-peptide reaction. A measure of 0.07% of propionaldehyde was estimated to be present in tripropylamine, which is lower than the overall impurity limit claimed by the manufacturer, but clearly enough to modify the proteins and peptides and generate artifacts that could complicate proteomics experiments. It is unclear why aldehydes are present in trialkylamines. They may be formed as a byproduct during the initial synthesis of the trialkylamine. Alternatively, it is possible that they were produced by air oxidation. Based on the results with trialkylamines and water, it is also possible that water facilitates this transformation very slowly. Therefore, when using trimethylamine, triethylamine or tripropylamine in biochemical studies, it should be considered that these modifications can affect the sample. These effects can be eliminated, if necessary, by mixing a primary amine with the trialkylamine.

## Figures and Tables

**Figure 1 molecules-26-06481-f001:**
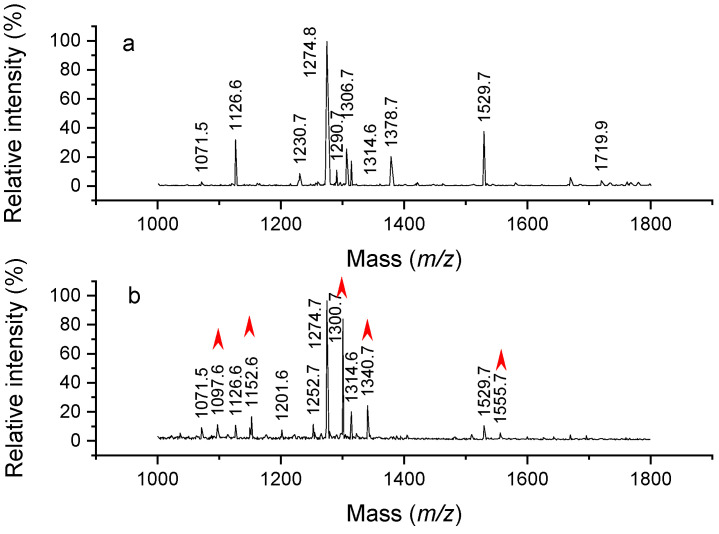
MALDI-TOF spectra of hemoglobin digest before (**a**) and after (**b**) adding triethylamine. Red arrows indicate 26 Da adducts.

**Figure 2 molecules-26-06481-f002:**
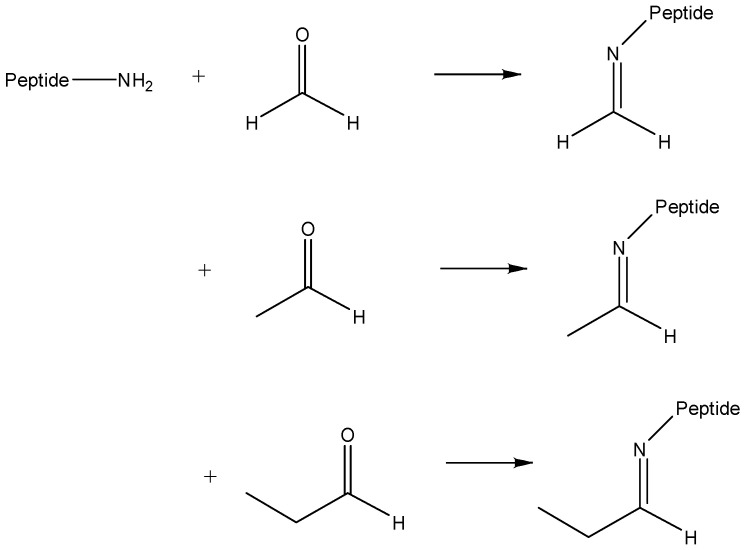
Schiff base reactions between peptides and three aldehydes leading to peptide mass shifts of 12, 26 and 40 Da.

**Figure 3 molecules-26-06481-f003:**
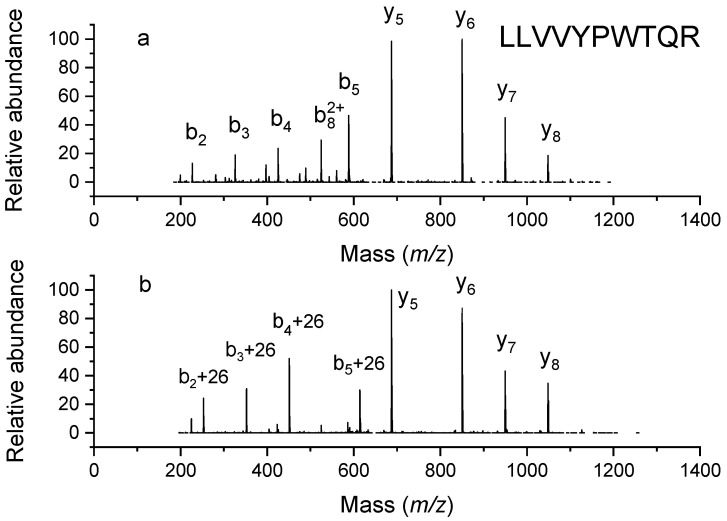
MSMS spectra of unlabeled LLVVYPWTQR (**a**) and its labeled counterpart (**b**).

**Figure 4 molecules-26-06481-f004:**
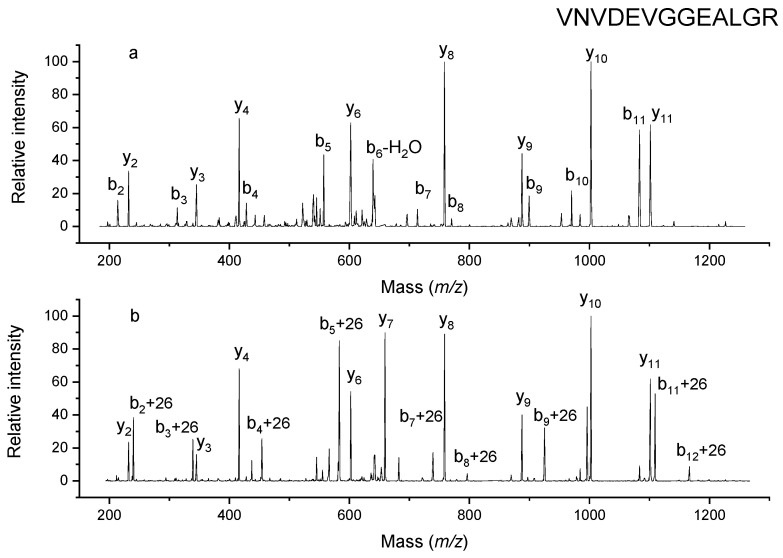
MSMS spectra of unlabeled VNVDEVGGEALGR (**a**) and labeled counterpart (**b**).

**Figure 5 molecules-26-06481-f005:**
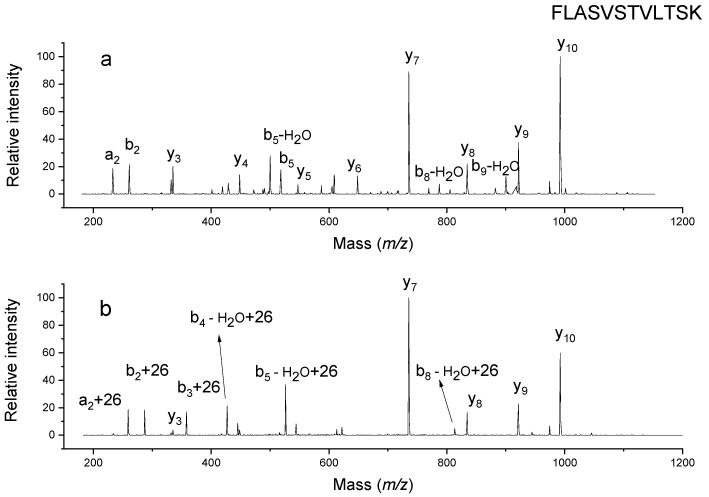
MSMS spectra of unlabeled FLASVSTVLTSK (**a**) and labeled counterpart (**b**).

**Figure 6 molecules-26-06481-f006:**
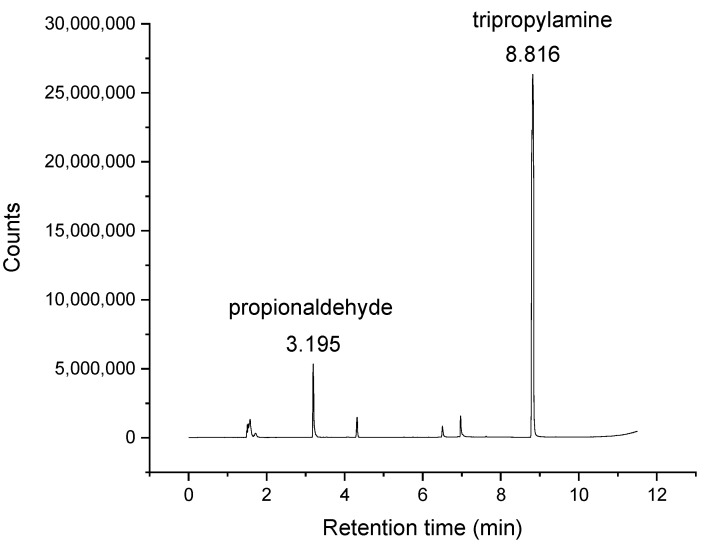
Total ion chromatogram of distilled tripropylamine from headspace analysis by GC-MS.

**Figure 7 molecules-26-06481-f007:**
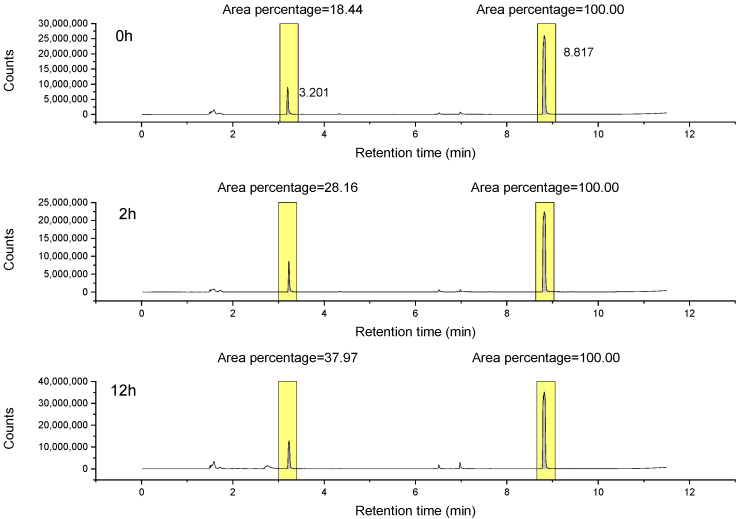
Total ion chromatograms of tripropylamine and water mixture after different incubation periods. The areas of propionaldehyde peaks were normalized to the areas of tripropylamine peaks in each chromatogram.

**Figure 8 molecules-26-06481-f008:**
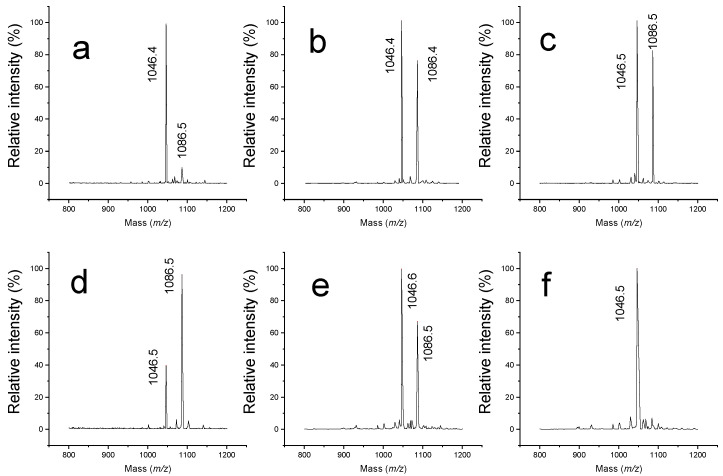
Mass spectra recorded the following reactions between angiotensin with: propionaldehyde in water (**a**); propionaldehyde in DMSO (**b**); propionaldehyde in pH 8 ammonium bicarbonate buffer (**c**); tripropylamine in water (**d**); tripropylamine in DMSO (**e**); propionaldehyde and propylamine in DMSO (**f**).

## Data Availability

All data interpreted in this study are contained within the article.

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
