# Peer review of "Aqueous Solutions of Peptides and Trialkylamines Lead to Unexpected Peptide Modification"

_molecules, 2021, doi:10.3390/molecules26216481_

Round 1

Reviewer 1 Report

Article is very good and interesting to read. There is just one small correction uL and uM should rather be μL and μM (micro symbol).

Author Response

We changed the u's to greek mu's as requested.

Reviewer 2 Report

Recommendation: Accept for publication in Molecules after minor revisions.

In this manuscript (Aqueous solutions of peptides and trialkylamines lead to unexpected peptide modification), Reilly and co-workers reported unexpected formation of adducts between trialkyl amines and buffered solutions of peptides. This study is interesting and timely.

The experimental design and results are convincing to demonstrate the formation of Schiff base products between low abundance aldehydes and amines. The authors conducted accurate mass measurements, headspace GC-MS and Schiff’s test with appropriate controls. This study is useful for researchers in the field of proteomics. So, I recommend this manuscript be accepted for publication after minor revisions. See some comments below.

  1. I would suggest the authors to provide chemical structures of the chemicals and adducts discussed in the manuscript. It can be added as a figure in the introduction. This will help the readers in better understanding of the results discussed here.

Author Response

We added a new Figure 2 that displays the aldehydes and adducts that they form, as requested.

Reviewer 3 Report

Reviewer comments

This manuscript describes “Aqueous solutions of peptides and trialkylamines lead to unexpected peptide modification”.  This is very interesting work on Schiff base products formation. When trialkyl amines are added to buffered solutions of peptides, unexpected Schiff base products can be formed. This work can help peptide and proteomics related experiments and its analysis using mass analysis. This manuscript is well written and properly referenced. This is useful work and can be consider for publication. Still, there are few shortcomings that will preclude its publication in the current form.

Major and minor concerns:

  1. Can author do more background search about source of aldehyde in trialkyl amines? If they found any relevant article for that, they can cite in background.
  2. Line 232, H NMR spectra should be corrected as 1H NMR
  3. Can authors provide some additional comments in background about source of these aldehydes in amines? It would be useful for researchers to understand and overcome this problem.
  4. Authors need to add more information in background information.

Manuscript can be considered for publication after these corrections.

Author Response

  1. We added a paragraph to the Introduction and sentences to the Conclusion that discuss the possible origin of the alddehydes either during the commercial synthesis or while in the lab.  Four references relating to this were added.
  2.  We changed H NMR to 1H NMR as requested.
  3.  See #1 above.
  4.  See #1 above.